# Pervaporation Membranes for Seawater Desalination Based on Geo–rGO–TiO_2_ Nanocomposites. Part 1: Microstructure Properties

**DOI:** 10.3390/membranes11120966

**Published:** 2021-12-08

**Authors:** Subaer Subaer, Hamzah Fansuri, Abdul Haris, Resky Irfanita, Imam Ramadhan, Yulprista Putri, Agung Setiawan

**Affiliations:** 1Material Physics Laboratory, Physics Department, Universitas Negeri Makassar (UNM), Makassar 90223, Indonesia; abd.haris@unm.ac.id (A.H.); misdayant25@gmail.com (M.); reskyirfanita@gmail.com (R.I.); ir221199@gmail.com (I.R.); yulpristaputri@gmail.com (Y.P.); 2Chemistry Department, Institut Teknologi Sepuluh November (ITS), Kampus ITS Sukolilo, Surabaya 60111, Indonesia; h.fansuri@chem.its.ac.id; 3Department of Metallurgical and Materials Engineering, Faculty of Engineering, Universitas Indonesia, Depok 16424, Indonesia; agung.setiawan72@ui.ac.id

**Keywords:** desalination, geopolymer, membrane, microstructure, pervaporation

## Abstract

This is the first of two papers about the synthesis and microstructure properties of the Geo–rGO–TiO_2_ ternary nanocomposite, which was designed to suit the criteria of a pervaporation membrane for seawater desalination. The performance and capability of Geo–rGO–TiO_2_ as a seawater desalination pervaporation membrane are described in the second paper. A geopolymer made from alkali-activated metakaolin was utilized as a binder for the rGO-TiO_2_ nanocomposite. A modified Hummer’s method was used to synthesize graphene oxide (GO), and a hydrothermal procedure on GO produced reduced graphene oxide (rGO). The adopted approach yielded high-quality GO and rGO, based on Raman spectra results. The nanolayered structure of GO and rGO is revealed by Transmission Electron Microscopy (TEM) images. The Geo–rGO–TiO_2_ ternary nanocomposite was created by dispersing rGO nanosheets and TiO_2_ nanoparticles into geopolymer paste and stirring it for several minutes. The mixture was then cured in a sealed mold at 70 °C for one hour. After being demolded, the materials were kept for 28 days before being characterized. Fourier Transform Infrared (FTIR) and X-ray Diffraction (XRD) measurements revealed that the geopolymer matrix efficiently bonded the rGO and TiO_2_, creating nanocomposites. Scanning Electron Microscopy (SEM) coupled with Energy Dispersive Spectroscopy (EDS) was used to examine the morphology of the outer layer and cross-sections of nanocomposites, and the results displayed that rGO were stacked on the surface as well as in the bulk of the geopolymer and will potentially function as nanochannels with a width of around 0.36 nm, while TiO_2_ NPs covered the majority of the geopolymer matrix, assisting in anti-biofouling of the membranes. The pores structure of the Geo–rGO–TiO_2_ were classified as micro–meso pores using the Brunauer–Emmet–Teller (BET) method, indicating that they are appropriate for use as pervaporation membranes. The mechanical strength of the membranes was found to be adequate to withstand high water pressure during the pervaporation process. The addition of rGO and TiO_2_ NPs was found to improve the hyropobicity of the Geo–rGO–TiO_2_ nanocomposite, preventing excessive seawater penetration into the membrane during the pervaporation process. The results of this study elucidate that the Geo–rGO–TiO_2_ nanocomposite has a lot of potential for application as a pervaporation membrane for seawater desalination because all of the initial components are widely available and inexpensive.

## 1. Introduction

The need for freshwater for drinking and for other beneficial uses is increasing alarmingly as the world population grows and industrialization in most big cities around the world grows rapidly. It is estimated that the number of people with limited access to freshwater in 2021 is about 670 million, and almost 2 billion of the world’s population depends on contaminated water [1,2,3].

Efforts to supply freshwater or drinkable water require new scientific and technological approaches and reasonable investment costs. Almost 98% of the water in the world is seawater, and hence, desalination is the best opportunity to expand the freshwater supply or relieve its shortage. Desalination is a complex process of removing dissolved salts from seawater to produce freshwater. Nowadays, seawater desalination all over the world is performed through multi-stage flash evaporation (MFE), multi-effect distillation (MED), and reverse osmosis (RO). These methods require high energy consumption and huge investment costs [2,4].

The reverse osmosis (RO) membrane has long been known as a means of supplying freshwater to people all over the world. The membrane is designed to have a high throughput and selectivity while maintaining mechanical integrity and biological fouling resistance. Producing membranes that meet all of these criteria is still difficult and time consuming. The separation efficiency of the membrane is limited by two factors: (i) the formation of specific species at the membrane surface, which causes polarization, and (ii) biofouling, which is caused by chemical solutions and membrane properties [5]. Some RO manufacturers have recently released new Sea Water Reverse Osmosis (SWRO) systems that comprise: (i) low fouling, (ii) improved boron rejection, and (iii) high permeability inorganic–organic membrane [4]. The anti-biofouling membrane, smooth, hydrophilic surfaces, and mechanical strength all require more investigation. A recent development of inorganic membranes, ceramics, and carbon-based membranes that match the requisite qualities and are easy to replicate has attracted a lot of interest [6].

The selection of materials and preparation processes for producing organic or inorganic membranes for seawater desalination has been the subject of extensive investigation. However, due to the inherent features of the material, the trade-off between membrane selectivity flux, biofouling deposition on the membrane surface, membrane durability and lifespan, and energy consumption, the majority of them do not meet the intended performance [6,7]. The goal of this research is to use inorganic nanocomposites comprised of geopolymer as a binder, reduced graphene oxide (rGO), and TiO_2_NPs as fillers to solve the aforementioned shortcomings of existing saltwater desalination.

Graphene is a two-dimensional material made up of single-layer carbon atoms arranged in a hexagonal lattice. The two-dimensional graphene sheets show remarkable electronic properties such as zero bandgap and effective mass. As a form of carbon, graphene is the leader of highly permeable and selective membranes which are suitable for fluid separation. This material is considered as the new type of RO membranes, having excellent properties such as strong, thin, robust, high flux, and energy-efficient separation [6]. The experimental attempt to produce nano porous graphene (NPG) is still inadequate since its production with reproducible quality and high mechanical strength is still difficult. Compared to graphene, graphene oxide (GO) or reduced Graphene Oxide (rGO) shows better prospects in membrane manufacture because of its simplicity and production in solution [8,9,10].

The usage of graphene-based material as a filler in membrane manufacturing with organic polymers such as polyimide and polylactic acid (PLA) as a binder has been recognized [11,12]. The organic binder was discovered to have excellent flowability, homogeneity, and to establish a strong chemical bond with GO or rGO, forming an excellent nanocomposite membrane. More recently, a nanofiltration membrane with outstanding thermomechanical properties and stable performance for water or acetone filtration was reported using rGO as a two-dimensional (2D) material combined with one-dimensional (1D) NaFe_2_S or NFS to form a ternary composite by using silkworm pupae protein as an organic binder [13].

Because of its large quantity and high potential for eliminating persistent organic contaminants, the nanocomposite rGO/TiO_2_ has been explored for wastewater purification. TiO_2_ is a semiconductor-based material with a band gap of roughly 3.24 eV that requires a photo-activation wavelength of less than 387 nm (UV) [14,15]. Due to quick recombination of electron–hole pairs, TiO_2_ photocatalyst was shown to have limited adsorption capacity for hydrophobic contaminants and low photonic efficacies during water treatment. The addition of graphitic materials to TiO_2_ increased its photocatalytic efficiency, which was influenced by an increase in specific surface area and charge carrier mobility. Because of their outstanding conduction magnitude, as well as their mechanical and chemical qualities, graphene and reduced graphene oxide have been used to layer TiO_2_. A one-step hydrothermal approach can be used to synthesize the rGO/TiO_2_ nanocomposite [16,17,18]. The anti-fouling properties of TiO_2_NPs could also be obtained from organic polymers such as polydopamine (PDA) coated on a nanocomposite of polybenzimidazole (PBI)-GO membrane, as well as a mixed membrane of cellulose acetate-polydopamine-sulfobetaine methacrylate (P(DA-SBMA)) nanoparticle for oil-in-water separation [19,20]. It has been reported that a combination of GO and PDA works well as a membrane anti-fouling agent.

Around the 1980s, Davidovits invented an amorphous to semi-crystalline aluminosilicate inorganic polymer or geopolymer. The importance of these minerals is particularly based on their natural availability in Si–Al minerals such as kaolinite (Al_2_Si_2_O_5_(OH)_4_). The empirical formula of geopolymer is written as M_n_ [(-SiO_2_)_z_–AlO_2_]_n_. w H_2_O, where Mn is a cation (the alkaline element), n is a degree of polycondensation, w ≤ 3 and z is 1, 2, or 3. The structures of geopolymers are zeolite-like, but their mechanical strength is different. Geopolymers offer a lot of applications, whether used pure, with aggregate or reinforced, due to their excellent mechanical properties, heat and fire resistance, and chemical stability. Traditionally, these applications can be divided into two categories: (i) structural products such as reinforced cement and concrete substitutes, and (ii) solidification technology for heavy metals and radioactive waste control. It is expected that geopolymers will become part of advanced materials for various applications with the inclusion of other materials such as TiO_2_, ZnO, CuO, Ag NPs, and graphene [21,22].

GO and rGO have been introduced into geopolymer paste in recent years to improve its characteristics and provide self-sensing capabilities to the geopolymer. Because of its large specific surface area, ultrahigh strength, and flexibility, GO is projected to provide desired qualities for manufacturing excellent geopolymer composites [23,24]. The incorporation of rGO/TiO_2_ into the geopolymer matrix is presented as a functional membrane for seawater desalination using the pervaporation method.

Pervaporative desalination has recently gained a lot of attention as a unique process for treating seawater [7,8,17,25,26,27]. To maintain a chemical potential difference for species migration over the other side of the membrane, the feed section of the membrane is exposed to preheated liquid while the permeate side is kept at a vacuum or cleansed with an inert gas in pervaporation. The diffusion concept is used to separate species flowing through a rigid membrane. Water molecules are drawn into the surface of a hydrophilic membrane, followed by movement across the membrane, and eventually evaporate on the other side of the membrane, according to the pervaporation mechanism.

Membrane materials with excellent properties such as hydrophilicity, heat resistance, and mechanical strength, such as geopolymer, reduced graphene oxide (rGO), and TiO_2_ in the form of nanocomposite, are suitable candidates for use as pervaporation membranes to achieve excellent pervaporation performance [25,28]. There has not been much investigation into the usage of geopolymer as a pervaporation membrane yet. He et al. [29] employed a hydrothermal technique to convert geopolymer into a self-supporting zeolite-NaA pervaporation membrane for seawater desalination with a membrane thickness of 9.4 mm and achieved a sodium ion rejection of 99.5 percent. Xu et al. [30] used geopolymer-gel-thermal-conversion (GGTC) to generate non-hydrothermal NaA zeolite and used a stainless steel-supported NaA zeolite membrane in the pervaporation (PV) of an ethanol–water mixture. Subaer et al. [31] developed a pervaporation membrane based on laterite zeolite–geopolymer for ethanol–water separation. This article describes the synthesis and the microstructure of Geo–rGO–TiO_2_ ternary nanocomposite as a potential pervaporation membrane for seawater desalination.

## 2. Materials and Methods

### 2.1. Synthesis of Geopolymer Paste

In this study, kaolin was used as a precursor of geopolymer due to its purity as an aluminosilicate mineral compared to fly ash or furnace slag. Kaolin was dehydroxylated at 750 °C for 4 h to transform it into a metakaolin phase in order to enhance its reactivity in an alkaline environment. The geopolymer paste was synthesized through the alkali-activation of metakaolin at a curing temperature of 60 °C for 2 h. The alkali solution was prepared by mixing NaOH, H_2_O and Na_2_SiO_3_. The mixture was left overnight before use. The elemental compositions of the starting materials were adjusted to have an atomic ratio of Si:Al = 1.5, Na:Al = 0.6, and H_2_O:Na_2_O = 10. These compositions were discovered to produce a Na-Poly (sialate-siloxo) type of geopolymer with high compressive strength and resistance to high temperatures and chemical attack [32]. The physico-chemistry properties of kaolin and metakaolin and the resulting geopolymers were measured by means of TG/DTA, FTIR, XRD, and SEM-EDS.

### 2.2. Synthesis of Reduced Graphene Oxide (rGO)

The rGO was produced through a hydrothermal reduction of graphene oxide (GO) in an autoclave at a temperature of 160 °C for 6 h. The GO was synthesized from graphite by using a modified Hummer’s method. The synthesis of GO was conducted by mixing 1 g of graphite with 27 mL of H_2_SO_4_ and 3 mL of H_3_PO_3_ solutions. Three grams of KMnO_4_ were added into the mixture under a magnetic stirrer at a speed of 600 rpm for 3 h. To complete the oxidation reaction, the mixture was poured into 200 mL of deionized water and 3 mL of 30% H_2_O_2_ was added. The GO suspension was then centrifuged for 4 h at the speed of 500 rpm. The resulting GO was washed by using 20% HCl, acetone, and deionized water several times until the pH reached 7. The microstructure properties of graphite, GO, and rGO were characterized using FTIR, XRD, Raman Spectroscopy, and TEM.

### 2.3. Synthesis of Geo–rGO–TiO_2_ Nanocomposite

Table 1 shows the compositions of the starting material for Geo–rGO–TiO_2_ nanocomposite. In this experiment, the mass of TiO_2_ was held constant at roughly 10% of the mass of metakaolin. The TiO_2_ NPs used in this study were supplied by Sanno, Indonesia, with an average particle size of 20 nm, and they were used without further treatment. Geo-GO-TiO_2_ ternary nanocomposite was produced by dispensing rGO and TiO_2_ nanoparticles into geopolymer paste and physically stirring the mixture for a few minutes until it appeared homogeneous. The mixture was then cured in a sealed mold at 70 °C for 1 h. The materials were stored for 28 days after being demolded before being characterized by means of FTIR, XRD, SEM-EDS, BET, and Splitting Tensile measurement.

The nanocomposites of Geo–rGO–TiO_2_ were fabricated to work as a pervaporation membrane with a dimension shown in Figure 1 for unsupported and supported membranes with pores ceramics.

Table 2 shows that adding TiO_2_ NPs lowers the composite’s density, which declines even more when more rGO is added. The addition of more pores and multi-layers of rGO that act as air channels has resulted in a decrease in sample density.

## 3. Results and Discussion

### 3.1. The Properties of Geopolymer Paste

Dehydroxylation of kaolin at 750 °C for 4 h yielded metakaolin, which was employed in the manufacturing of geopolymer. TG-DTA was used to identify the exact temperature at which kaolin dehydroxylation occurs, as shown in Figure 2a. The DTA trace is typical of kaolinite, with a dehydroxylation endotherm around 541 °C and an exotherm about 984 °C, indicating spinel production. The weight loss at 450 °C can be attributed to a pre-dehydration process produced by restructuring in the kaolin octahedral layer, as seen by the TG curve. At 450 °C, the loss of OH lattice water begins, and dehydroxylation is essentially complete at 650 °C. This result is consistent with other forms of kaolin that have been dehydroxylated [33,34]. The loss of lattice water causes the crystalline structure of kaolinite to collapse, changing the coordination number of Al from VI to IV [35].

Figure 2b shows the XRD pattern of kaolin used in this study and all the observed reflections match those from ICDD 01-089-6538. The diffractogram of metakaolin indicates that all the kaolin reflections have been eliminated, leaving an amorphous pattern with ancillary peaks due to quartz (ICDD 033-065-0466), silicon oxide (SiO_2_) (ICDD 01075-3901), magnesium oxide (MgO) (ICDD 01075-9569), and calcium oxide (CaO) (ICDD 01076-8925).

Figure 3a,b shows the FE-SEM image of kaolin and metakaolin, respectively. The SEM image of metakaolin shows that the loss of water during dehydroxylation of kaolinite destroys dioctahedral kaolin and transforms the octahedrally coordinated Al layer into the tetrahedral form, but the Si–O sheets of the kaolin structure are largely conserved [33]. The EDS result of metakaolin showed that Si = 22.7 wt.%, and Al = 22.0 wt.%.

The geopolymer paste produced in this study was categorized as Na-Poly (sialate-siloxo) type, with a molar ratio of the starting materials of Si: Al = 1.5 and Na: Al = 0.6. The morphology of the geopolymer surface was observed by using FlexSEM1000 coupled with EDS for elemental compositions examinations, as shown in Figure 4.

The SEM image of the geopolymer surface shows the relatively homogenous geopolymer matrix with visible apparent micropores. The atomic ratios of Si:Al = 1.78 and Na:Al = 0.62 are determined by EDS. The calculated Si:Al and Na:Al of the starting materials were slightly greater than these magnitudes. This is normally caused by the high amount of oxide minerals as an impurity in metakaolin, particularly SiO_2_ as well as the high modulus of SiO_2_/Na_2_O of sodium silicate. The synthesis of geopolymer involves the dissolution of mineral aluminosilicates (such as metakaolin, fly ash, and slag furnace) in an alkaline solution, hydrolysis of Al^3+^ and Si^4+^ components, and condensation of specific aluminate and silicate species. In alkaline conditions, the resultant of Al atom is determined by aluminate species of [Al(OH)_4_]^−^, whereas Si atom is determined by silicate species of [SiO(OH)_3_]^−^ and [SiO_2_(OH)_2_]^−^ to [SiO(OH)_3_]^−^ increasing with pH value. The setting and hardening of geopolymer take place as a result of condensation between aluminate and silicate species. The experimental results, such as the one reported by Weng and Sagoe-Crentsil [36], indicate that the condensation between aluminate and silicate species occurs more rapidly than that between silicate species themselves, but the exact mechanism involved is not fully understood yet. This condition, together with the presence of impurity minerals in the starting materials, will result in a minor variation from the calculated compositions of the geopolymer final product.

### 3.2. The Properties of rGO

Figure 5a shows the diffractogram of graphite used, as well as GO and rGO produced in this study. The diffractogram illustrates the shift of planes (002) and (001) from graphite to GO to rGO. Plane (002) is at 2θ = 9.42° and plane (001) is at 2θ = 42.29° for GO. Plane (002) is at 2θ = 23.52° (d-spacing = 0.38 nm) for rGO, and plane (001) is at 2θ = 42.95° (d-spacing = 0.21 nm) for rGO. The rearrangement of carbon atoms during oxygen reduction from GO to rGO is well recognized to induce these shifts. Other researchers have observed similar findings [37,38,39].

The Raman spectra of GO and rGO are shown in Figure 5b. The peak of the D band in the Raman spectra of GO appears to be caused by sp^3^ defects at 1350 cm^−1^ (ID = 944.123 counts), while the peak of the G band at 1590 cm^−1^ (IG = 1021.69 counts) appears to be caused by in-plane vibrations of sp^2^ carbon atoms. The peak of the D band of rGO is at 1352 cm^−1^ (ID = 1617.45 counts), while the peak of the G band is at 1587 cm^−1^ (IG = 1333.78 counts). The shift of G band peaks at rGO is linked to the property of self-healing, which preserves the hexagonal network of carbon atoms [40]. The ID/IG ratio increased from 0.92 to 1.21 when GO was transformed to rGO, indicating that structural disorder had increased.

Figure 6 displays SEM image and FEI Tecnai 200 kV TEM image rGO, which show the material has a layered structure and are made up of several crumpled nanosheets with a clattered structure of roughly 100–200 nm in size. The images also show that rGO exhibits closely packed plate structure and a clean surface.

The rGO was extensively combined with geopolymer paste during the synthesis of Geo–rGO–TiO_2_, and it was discovered to stack on the surface as well as the bulk of the geopolymer. The rGO layer will function as nanochannels, allowing water molecules to freely travel while salt is rejected. Because rGO nanochannels are larger than water molecules but smaller than Na^+^ and Cl^−^ ions, this is achievable [10].

### 3.3. The Properties of TiO_2_ NPs

The FTIR and XRD of TiO_2_ NPs are shown in Figure 7. Functional group of TiO_2_ is observed at around 1623 cm^−1^ (Ti-OH vibration) and at 483 cm^−1^ (Ti-O vibration). The broad band around 3404 is O–H stretching vibration [41]. Diffractogram of TiO_2_ NPs shows that the phase of TiO_2_ was anatase (ICDD 01-075-2552). The average of crystal size of TiO_2_ NPs was calculated by using Debye–Scherrer formula,
(1)$$D=\frac{k\;\lambda }{\beta \cos\theta \;}\;$$
*k* = 0.9, *λ* = 0.15406 nm, *β* = FWHM, *θ* = peak position. It was found that the average crystal size of TiO_2_ NPs used in this study is 16.47 nm.

### 3.4. The Structure and Morphology of Geo–rGO–TiO_2_ Nanocomposites (Pervaporation Membranes)

Figure 8 shows the spectrum of the FTIR sample Geo–rGO–TiO_2_. The spectrum reveals prominent peaks around 3453 and 1642 cm^−1^, which are characterized as stretching and bending vibrations of (OH)^−^, respectively. The asymmetric stretching of the O–C–O bond is confirmed by the peak at 1386 cm^−1^ [42]. The asymmetrical stretching vibration of the Si–O–Si bond is observed at about 1014 cm^−1^. The band around 594 cm^−1^ is attributed to Al–O–Si stretching vibrations, and the bending vibration of the Si–O–Si bond is observed at 445 cm^−1^ [42,43]. A peak at 694 cm^−1^ is recognized as the bending vibration of Al (IV)–O–Si in a cyclic structure of geopolymer [44]. The functional group of Ti–O–Ti vibration is observed at 1379 and 561 cm^−1^ Ti–O bonds in the TiO_2_ lattice [45]. These wavenumbers are slightly shifted to a higher wavelength due to the presence of rGO in the geopolymer network.

Furthermore, rGO absorption bands can be observed at 1639 cm^−1^ attributed to C=C bending vibration, 1384 cm^−1^ and 2145 cm^−1^ are due to C–O bending vibration, and the band at 1404 cm^−1^ can be attributed to the C–H vibration band [46,47]. The existence of TiO_2_ and rGO absorption bands is an indicator of the successful inclusion of rGO and TiO_2_ in the network of geopolymer.

The Diffractograms of the geopolymer and Geo–rGO–TiO_2_ nanocomposites (Figure 9a) show that the amorphous hump of the geopolymer shifts slightly to lower 2θ as the rGO-TiO_2_ content increases. This is related to the rearrangement of the aluminosilicate network of geopolymer [32]. The phase of NaAl_2_(AlSi_3_)O_11_ is present in all samples and forms a highly crystalline phase on the surface of Geo–rGO–TiO_2_ (1, 3), as shown in Figure 10d. This crystalline phase might be triggered by the presence of a high amount of TiO_2_ and rGO and be responsible for the attainment of the high mechanical strength of the sample as measured by means of a tensile splitting test [48]. The distribution of the chemical compositions (wt.%) of the sample phases is shown in Figure 9b. The highest composition of NaAl_2_(AlSi_3_)O_11_ is in the sample Geo–TiO_2_ (3).

The microstructure of the membranes was examined by using SEM and their elemental compositions were examined by means of EDS. Figure 10 show the morphology of membrane’s outer layers. The images were taken at two different magnifications for each sample, 500 and 8000×, to expose the majority of the membrane surface as well as more detailed morphology.

SEM images of the outer layer for all samples showed that the geopolymer matrix is reasonably homogeneous, indicating good polymerization of metakaolin. The existence of unreacted metakaolin can be seen in the sample geopolymer, as indicated by the red circle (Figure 10a,a’). The existence of unreacted metakaolin in the geopolymer matrix has a substantial impact on the mechanical strength, pore distribution, and size of the resultant nanocomposite [32,49], and hence the overall quality of the Geo–rGO–TiO_2_ nanocomposite as a membrane. Nanoparticle TiO_2_ is randomly distributed in samples (b and b’) and (c and c’). In sample (c and c’), the nanosheet of rGO along with TiO_2_ nanoparticles covers some parts of the geopolymer surface. As shown in figures (d and d’), the geopolymer matrix in the sample Geo–rGO–TiO_2_ (1, 3) becomes highly crystalline and porous in sample Geo–rGO–TiO_2_ (1, 3) due to the formation of NaAl_2_(AlSi_3_)O_11_ as confirmed by XRD. The quantity of Na atomic percent in this sample is very high (28.82 at.%), as confirmed by the EDS result (Table 3). Similar crystal formation of geopolymer paste is also observed by Saafi et al. [23] and Yan et al. [24] for rGO–geopolymer nanocomposite, and by Wan et al. [50] for incorporating ZnO in their geopolymers.

The elemental compositions of the outer layers based on EDS measurements for the whole image fields of Figure 10 are shown in Table 3. The atomic ratio of Ti and C atoms in the Geo–TiO_2_ and Geo–rGO–TiO_2_ samples indicates that the particles of TiO_2_ and rGO are evenly distributed and bound in the matrix of the geopolymer, forming good composites.

Figure 11a,b show the morphology of cross-sections of each membrane geopolymer samples, taken at two difference magnifications, 500 and 8000×, respectively, in order to inspect the morphology of the geopolymer paste as well as the morphology of Geo–rGO–TiO_2_ nanocomposite. The elemental compositions of the cross-section of each specimen are shown in Table 4. The morphology of metakaolin remained visible after geopolymerization took place. In samples (c), the rGO nanosheets and TiO_2_ nanoparticles cover a lot of areas. The distribution of rGO and TiO_2_NPs on the surface and cross-sections of geopolymer is one of the prerequisites if the composite is to be applied as a pervaporation membrane in the process of seawater desalination, with rGO serving as nanochannels for water molecules and TiO_2_NPs acting as an anti-fouling material [9,10,17].

The microstructure of Geo–rGO–TiO_2_ demonstrates that rGO may be manufactured as thin films stacked inside the surface and the bulk of geopolymer, forming 2D nanochannels of two neighboring rGO that also function as desalination membranes. These nanochannels have a width of roughly 0.74 nm for GO, 0.36 nm for rGO, and 0.68 nm for graphene [51], which is substantially greater than water molecules (0.275 nm) and so can be penetrated by water. On the other hand, by using size exclusion and Columbic interactions, some ions such as Na^+^ (0.716 nm), K^+^ (0.662 nm), Mg^2+^ (0.856 nm), and Cl^−^ (0.664 nm) can be detached from seawater, allowing for desalination via the rGO membrane [10]. Furthermore, TiO_2_ nanoparticles will help to improve the durability and prevent deposition of any species on the surface of the membrane [45,52].

### 3.5. The Surface Area and Porosity of Geo–rGO–TiO_2_ Nanocomposites (Pervaporation Membranes)

The Brunauer–Emmett–Teller (BET) nitrogen adsorption method was performed to study pore structure, the surface area, as well as the size and distribution of porosity. In the BET method, adsorption is performed over a range of relative pressure (*P*/*P*_0_) by the volumetric dosing of N_2_ onto the sample at 77 K. The BET equation can be written as [53,54];
(2)$$\frac{1}{W\;\left[\left(\frac{P_{0}}{P}\right)-1\right]}=\frac{1}{W_{m}\;C}+\frac{C-1}{W_{m}\;C}\left(\frac{P}{P_{0}}\right)$$
where *W* = weight of gas adsorbed, *P/P*_0_ = relative pressure, *W_m_* = weight of adsorbate as monolayer, and *C* = BET constant.

The Equation (2) is applied to the isotherm data to produce BET multi-point plot to determine Wm and surface area (*S*). The multi-plot of BET function against *P/P*_0_ is written as;
(3)$$\frac{1}{\left[W\left(\frac{P}{P_{0}}\right)-1\right]}\;\mathrm{vs}.\;\frac{P}{P_{0}}$$
and the surface area (*S*) is given by;
(4)$$S=\frac{W_{m\;}N\;A_{cs}}{M}$$
where *N* = Avogadro’s number, *A_cs_* = Adsorbate cross-sectional area, and *M* = molecular weight of adsorbate.

Figure 12 and Figure 13 show the isotherm graphs of geopolymer and Geo–rGO–TiO_2_ (1, 3), respectively. The isotherm graphs reveal the type of BET for materials having micro–meso pores in the range of 1.5 to 100 nm. The graphs are also a source for calculating the surface area, the average size, and distribution of porosity. Geopolymer is a porous cementitious material with a wide range of pore sizes and other voids, such as entrapped and entrained air and capillary pores. When a geopolymer’s surface is exposed to water, it absorbs most of the water due to its high hydrophilicity. This reduces the material’s lifetime and causes a problem with durability. As a result, to function as a good membrane, the geopolymer surface must be water resistant. If the geopolymer’s hydrophobicity is increased, the rate of water penetration through the material will be slowed.

Pore distributions for geopolymer and Geo–rGO–TiO_2_ (1, 3) samples calculated by using BJH (Barrett–Joyner–Halenda) method are shown in Figure 14a,b, respectively.

Using N_2_ adsorption data, Figure 14a,b demonstrate the pore size distribution as well as the differential of pore volume as a function of pore size using the Barret–Joyner–Halenda (BJH) method. The geopolymer sample has pore sizes ranging from 15 to 1345 Å, with a maximum pore volume of 8.94 × 10^−4^ cc g^−1^ (Max 1) and 3.61 × 10^−4^ cc g^−1^ (Max 2). Geo–rGO–TiO_2_ (1, 3) pores range in size from 15 to 1279 Å, with a maximum pore volume of 9.457 × 10^−4^ cc g^1^ (Max 1) and an average of 5.92 × 10^−4^ cc g^1^ (Max 2). The introduction of rGO and TiO_2_ nanoparticles results in changes in size distribution and volume. Three BET parameters for the membranes generated in this study are listed in Table 5.

Table 4 shows that Geo–TiO_2_ (3) has the highest pore surface area and the smallest pore diameter. With an average pore diameter of roughly 12.90 nm, sample Geo–rGO–TiO_2_ (0.5, 3) has the second greatest pore surface area. As seen in sample Geo–rGO–TiO_2_ (1, 3), adding additional rGO reduces pore surface area while increasing pore diameter. The total pore volume for Geo–rGO–TiO_2_ is relatively the same. All BET parameters are the main factors influencing the permeate flux and selectivity of the resulting membranes. The results of BET measurements will become a major consideration in refining the microstructure of Geo–rGO–TiO_2_ nanocomposite that fulfills the requirement as a desalination membrane.

Because the average pore size of the Geo–rGO–TiO_2_ nanocomposite formed is roughly 13 nm, the percentage of salt rejection will be quite low. The preliminary test conducted to examine the ability of the prepared membranes to reject salt during the pervaporation process revealed that at 60 °C feed temperature, the drop in salinity of saltwater with an initial salinity of 28.6 ppt was only about 30 percent, with a flux of around 2.75 kg m^−2^ h^−2^. The results suggest that the selectivity of the membrane should be improved by reducing the pore size and total pore volume of the Geo–rGO–TiO_2_ nanocomposite. This can be achieved by solidifying the geopolymer paste through longer curing time [49], and increasing the weight percentage of rGO and TiO_2_NPs filler.

In a pervaporation process of seawater, the feed side of the membrane is directly contacted with the saturated liquid while the permeate side is kept at a vacuum to maintain the chemical potential difference for water to diffuse into nanopores across the membrane thickness, similar to the transport mechanism of a nanofiltration membrane. In this case, the pervaporation membrane’s mechanism is a phase change process, and unlike reverse osmosis, it does not need to overcome the osmotic pressure of salt water [55,56].

### 3.6. The Splitting Tensile Result of Geo–rGO–TiO_2_ Nanocomposites (Pervaporation Membranes)

One of the most significant qualities of the pervaporation membrane is mechanical strength, which allows it to endure the high pressure of hot liquid that is constantly striking its surface. The mechanical strength of the samples in Figure 1i was evaluated using a Tokyo Testing Machine (TTM) for splitting tensile measurements. The horizontal magnitude of the sample’s splitting tensile strength (*P*, MPa) was measured using the equation.
(5)$$P=\frac{2\times F}{d\times L}$$
where *F* = the ultimate applied force until the sample break (N), *d* = the thickness of the sample, and *L* = the length of the sample.

Figure 15 shows that the inclusion of TiO_2_ nanoparticles and rGO enhances the average magnitude of splitting tensile strength. When the standard deviation of three specimens is taken into consideration, the samples Geo–rGO–TiO_2_ (0.5, 3) and Geo–rGO–TiO_2_ (1, 3) had the maximum strength. The magnitude of the splitting tensile strength of the geopolymer produced in this study is slightly lower than that of a silica fume-based geopolymer reported by Daniel et al. [57].

Based on the preliminary investigation, the strength of these membranes was able to withstand seawater pressure as high as 3.0 Pa inside the pervaporation membrane house at temperatures up to 80 °C.

### 3.7. Contact Angle Measurement

Geopolymer, like zeolite, is a porous material with a high hydrophilicity due to the unsaturated-OH and O-functional groups in its network [58,59]. Nanoparticles such as SiO_2_, TiO_2_, and graphene-based materials can be used to modify the porosity of geopolymers [60,61,62]. The addition of TiO_2_NPs in particular was found to produce a self-cleaning and anti-bacteria geopolymer [45,52]. Carbon nanostructure such as carbon nanotube, carbon nanofiber, fullerene, rGO, and graphene shows excellent hydrophobia properties and becomes widely used as a coating material [63,64]. The addition of rGO-TiO_2_NPs into geopolymer paste was expected to improve the microstructure of nanocomposite which fulfills the requirement to be applied as a pervaporation membrane [16,23].

The angle of water contact with the solid surface is used to identify a material’s hydrophilicity or hydrophobicity. Figure 16 shows the water contact angle of Geo–rGO–TiO_2_ nanocomposites measured at three separate spots on each specimen’s surface. It can be seen that the inclusion of TiO_2_NPs and rGO-TiO_2_NPs enhanced the contact angle of geopolymer significantly and hence improving its hydrophobicity. Similar results were reported by Ranjbar et al. [61] for fly-ash geopolymer added with graphene nanoplatelet. The increased hydrophobicity of the Geo–rGO–TiO_2_ nanocomposite is an important factor in preventing excessive seawater penetration into the membrane during the pervaporation process.

The hollow pervaporation membrane for seawater desalination fabricated by using geopolymer-based material was reported by He et al. [29]. The authors intentionally transformed the geopolymer into self-supporting zeolite as a membrane with a thickness of approximately 10 mm, a compressive strength of around 57 MPa, a micropore volume of around 0.049 cc g^−1^, and achieved 99.5% sodium ion rejection at a temperature of 90 °C.

The features of the geopolymer membranes described here are different. The Geo–rGO–TiO_2_ nanocomposite membranes produced in this study had an average micropore size of 0.1764 cc g^−1^, a thickness around 2.00 mm, and a splitting tensile strength of 0.35 MPa. The preliminary test of the produced membranes showed that the salt rejection was about 30% and the water flux was around 2.75 kg m^−2^ h^−2^ at a feed temperature of 60 °C. The results suggest that the membranes developed for desalination of saltwater are still ineffective. In order to function as an excellent seawater desalination pervaporation membrane, it is necessary to fine tune the geopolymer microstructure, pore size and total pore volume, membrane thickness, and the amount of rGO and TiO_2_ NPs included into the geopolymer network.

## 4. Conclusions

A ternary nanocomposite based on geopolymer, rGO, and TiO_2_ was successfully synthesized as a potential inorganic pervaporation membrane for seawater desalination. The physico-chemical properties of the fabricated metakaolin-based geopolymer, rGO, and TiO_2_ nanoparticles influence the microstructure and strength of the manufactured Geo–rGO–TiO_2_ nanocomposite. The addition of TiO_2_NPs and rGO initiated the crystalline phase of the geopolymer matrix, and enhanced the mechanical strength of the membranes. The highest tensile strength of the membrane is about 0.35 MPa and it is able to withstand the feed pressure of up to 3 Pa. The Geo–rGO–TiO_2_ nanocomposite roughly fulfilled all of the parameters for a pervaporation membrane, as measured by pore surface area, pore size, and distribution. The mean pore size was found to be around 13 nm, and the pore volume was found to be around 0.1764 cc g^−1^, both of which must be reduced for the nanocomposite to function as an effective membrane for seawater desalination. The addition of rGO and TiO_2_NPs to the geopolymer increases its hydrophobicity, resulting in a water contact angle of up to 63°, that would prevent excess seawater from permeating the membrane during the pervaporation process. The presence of rGO nanosheets on the surface and in the bulk of the geopolymer will also function as nanochannels, allowing water to freely flow out of the membrane while rejecting salt. The TiO_2_ NPs’ superior qualities will aid the geopolymer matrix in improving the membrane’s anti-biofouling properties. The results of this study suggest that Geo–rGO–TiO_2_ nanocomposite is a new functional inorganic material that has a lot of potential for application as a pervaporation membrane for seawater desalination because all of the initial components are widely available and inexpensive.

## Figures and Tables

**Figure 1 membranes-11-00966-f001:**
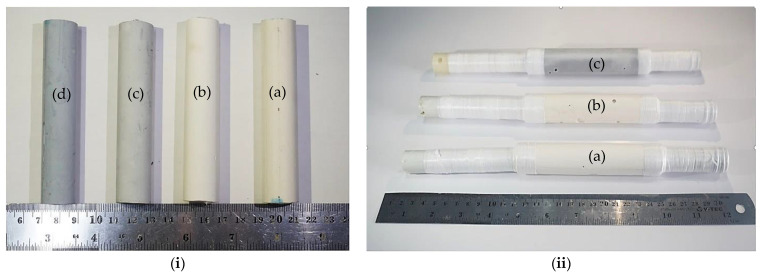
(**i**) Unsupported, and (**ii**) supported pervaporation membranes for desalination of seawater: (a) geopolymer; (b) Geo–TiO_2_ (3); (c) Geo–rGO–TiO_2_ (0.5, 3); (d) Geo–rGO–TiO_2_ (1, 3). All specimens were stored in the open air for 28 days prior to any characterization.

**Figure 2 membranes-11-00966-f002:**
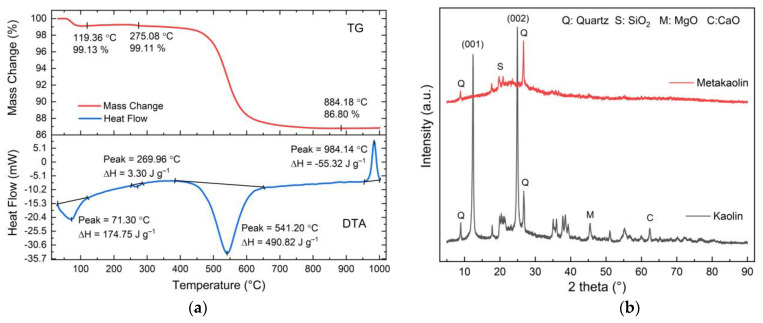
(**a**) TG/DTA of kaolin measured in the range of 25–1000 °C with a heating rate of 10 °C/min. The endotherm peak at 541 °C indicates the phase change of kaolin from crystalline state into an amorphous state, and (**b**) diffractogram of kaolin and metakaolin. Dehydroxylation at 750 °C changes the crystallinity of kaolin into an amorphous metakaolin.

**Figure 3 membranes-11-00966-f003:**
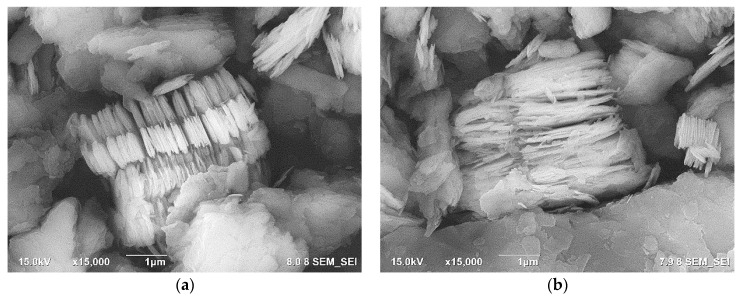
FE-SEM images (**a**) kaolin and (**b**) metakaolin. The images reveal the platy morphology of kaolin crystallinity, which is largely preserved in an amorphous metakaolin.

**Figure 4 membranes-11-00966-f004:**
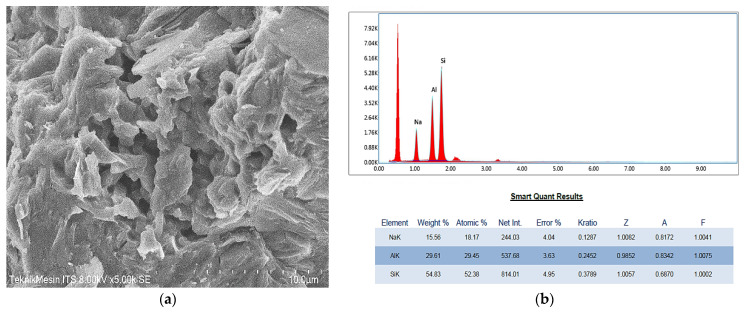
(**a**) Morphology of geopolymer surface and (**b**) EDS spectrum of geopolymer surface which was taken in the whole field area of the image.

**Figure 5 membranes-11-00966-f005:**
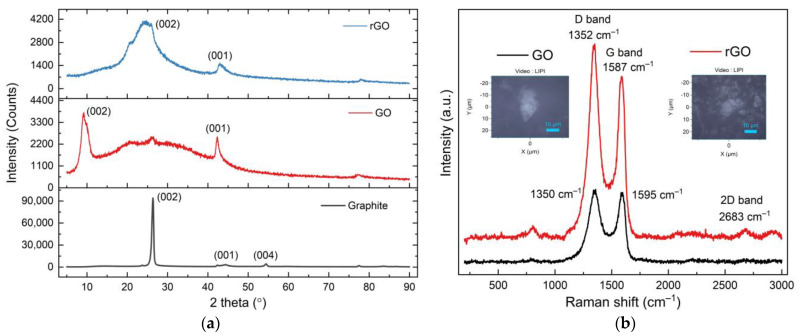
(**a**) Diffractogram of graphite, GO, and rGO. The transformation of graphite into GO and rGO shifted the planes (001), (002), and (004) and (**b**) Raman spectra of GO and rGO showing the peak locations of Dband and Gband.

**Figure 6 membranes-11-00966-f006:**
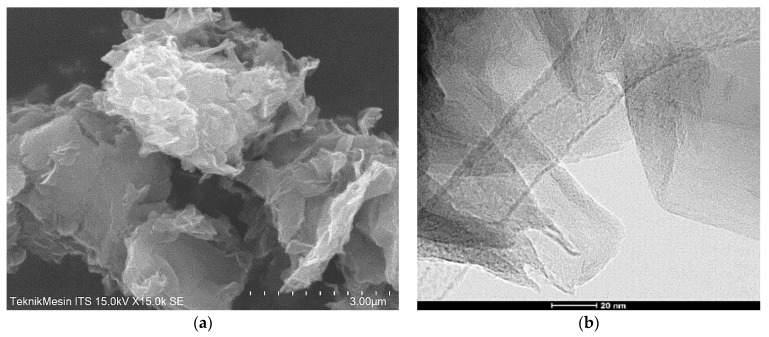
(**a**) SEM image, and (**b**) TEM image of rGO. The images show that the rGO surface reveals crumpled thin layers.

**Figure 7 membranes-11-00966-f007:**
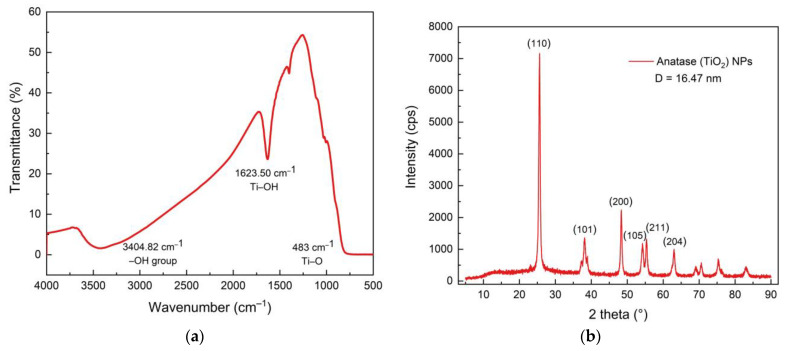
(**a**) FTIR spectra showing the vibration band of Ti-O at 483 cm^−1^, and (**b**) diffractogram of the anatase phase of TiO_2_ NPs used in this study.

**Figure 8 membranes-11-00966-f008:**
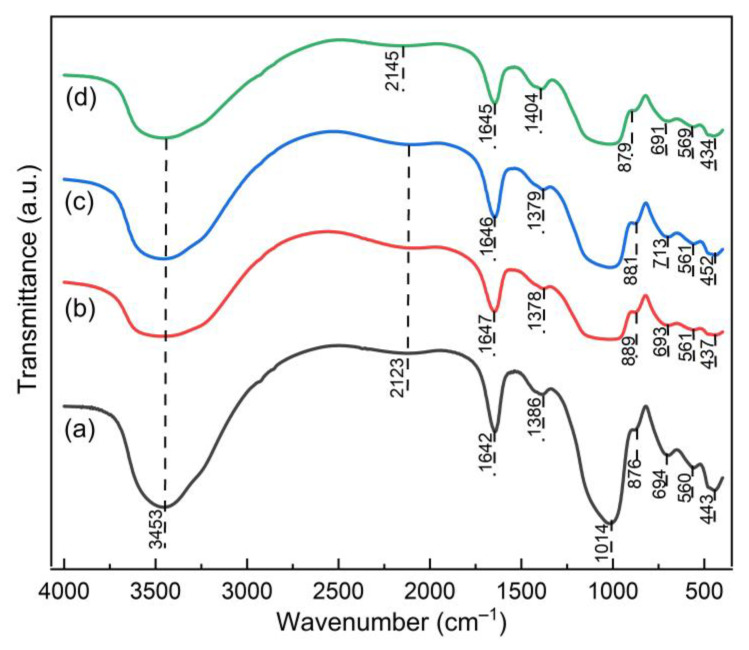
FTIR spectrum of pervaporation membranes shows the vibration bands of the functional groups formed in the network of (**a**) geopolymer; (**b**) Geo–TiO_2_ (3), (**c**) Geo–rGO–TiO_2_ (0.5, 3), (**d**) Geo–rGO–TiO_2_ (1, 3).

**Figure 9 membranes-11-00966-f009:**
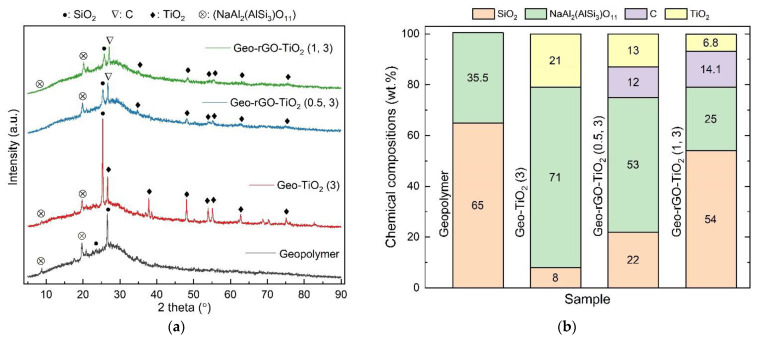
(**a**) Diffractogram, and (**b**) the distribution of chemical compositions of the membranes: Geopolymer, Geo–TiO_2_ (3), Geo–rGO–TiO_2_ (0.5, 3), Geo–rGO–TiO_2_ (1, 3). The addition of rGO and TiO_2_ did not change the amorphous nature of geopolymer.

**Figure 10 membranes-11-00966-f010:**
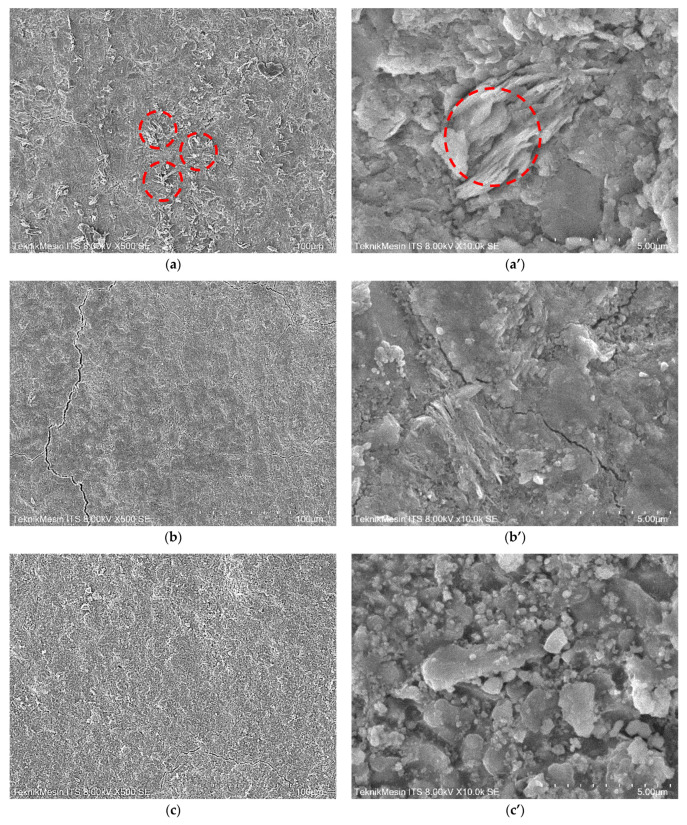
SEM images of the outer layers at two difference magnifications, 500× and 8000× for each sample: (**a**,**a’**) Geopolymer, (**b**,**b’**) Geo–TiO_2_ (3), (**c**,**c’**) Geo–rGO–TiO_2_ (0.5, 3), (**d**,**d’**) Geo–rGO–TiO_2_ (1, 3). Red circles in sample geopolymer indicate the unreacted metakaolin.

**Figure 11 membranes-11-00966-f011:**
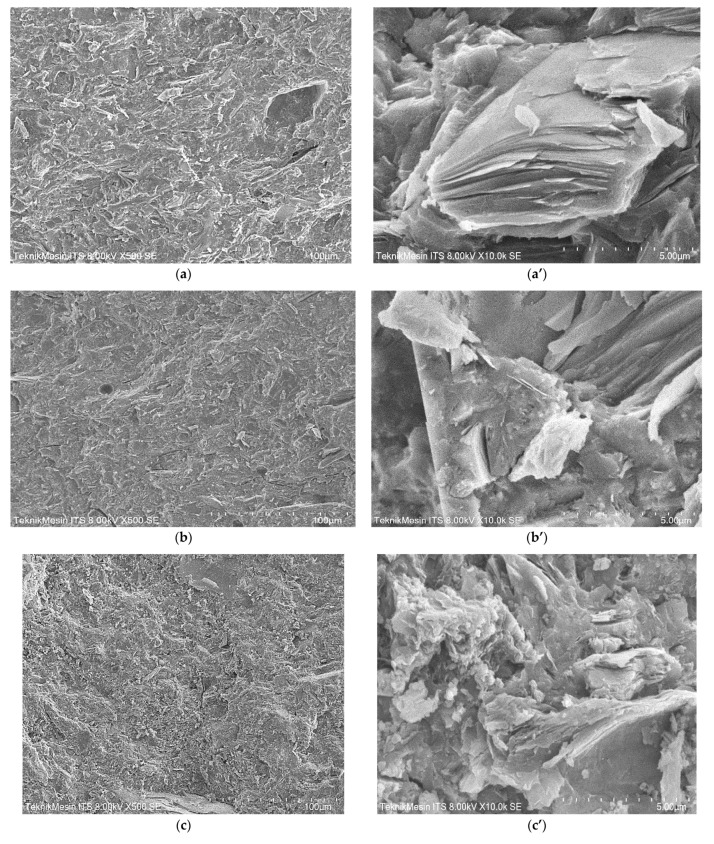
SEM images of the cross-sections at two different magnifications, 500 and 8000× for each sample: (**a**,**a’**) Geopolymer; (**b**,**b’**) Geo–TiO_2_ (3); (**c**,**c’**) Geo–rGO–TiO_2_ (0.5, 3); (**d**,**d’**) Geo–rGO–TiO_2_ (1, 3).

**Figure 12 membranes-11-00966-f012:**
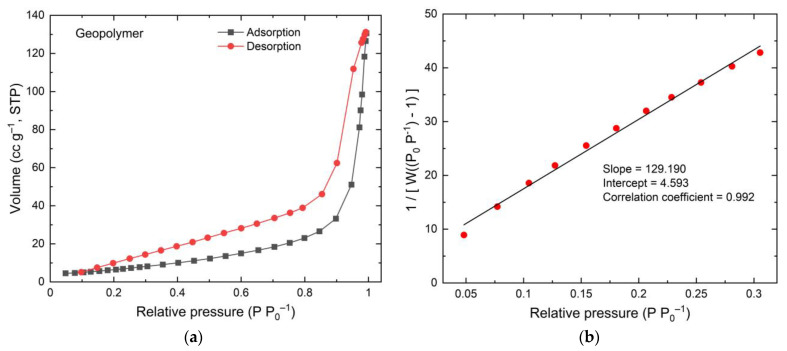
BET result of geopolymer (**a**) isotherm, and (**b**) multi-point BET.

**Figure 13 membranes-11-00966-f013:**
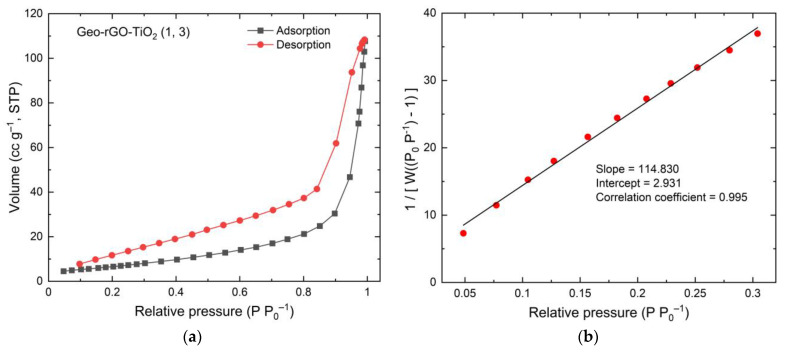
BET results of Geo–rGO–TiO_2_ (1, 3) nanocomposite (**a**) isotherm, and (**b**) multi-point BET.

**Figure 14 membranes-11-00966-f014:**
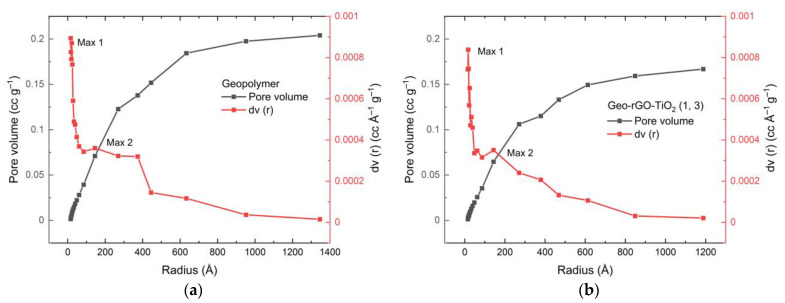
Pore distribution calculated by using BJH (Barrett–Joyner–Halenda) method for (**a**) geopolymer; (**b**) Geo–rGO–TiO_2_ (1, 3).

**Figure 15 membranes-11-00966-f015:**
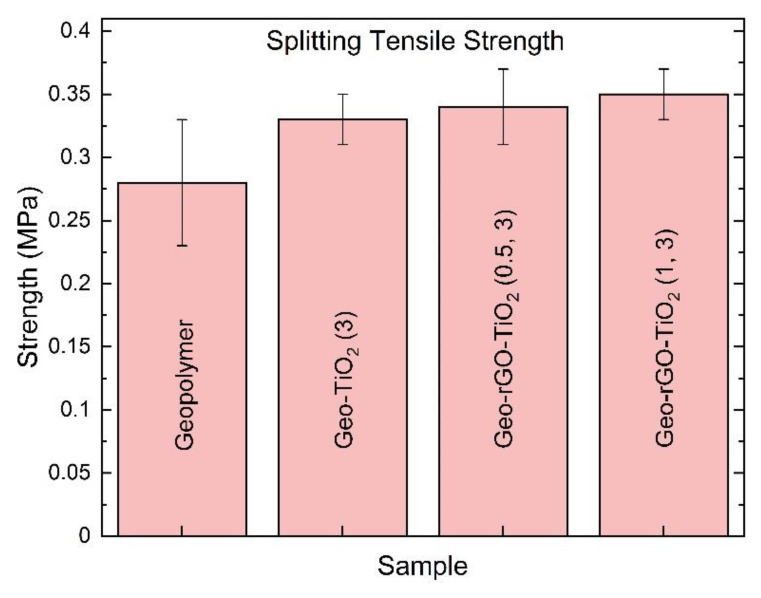
The magnitude of the splitting tensile strength of the unsupported pervaporation membranes. The error bars were taken as standard deviation of three specimens for each sample.

**Figure 16 membranes-11-00966-f016:**
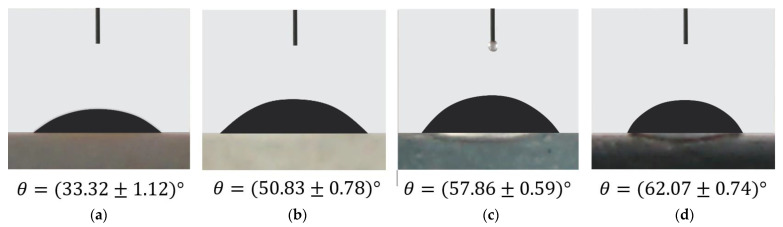
Water contact angle for sample: (**a**) geopolymer; (**b**) Geo–TiO_2_ (3); (**c**) Geo–rGO–TiO_2_ (0.5, 3), and (**d**) Geo–rGO–TiO_2_ (1, 3).

**Table 1 membranes-11-00966-t001:** Compositions of the starting materials for nanocomposite Geo–rGO–TiO_2_, in gram.

Sample	Metakaolin	NaOH	Na_2_SiO_3_	H_2_O	rGO	TiO_2_
Geopolymer	30	3	24	9	0	0
Geo–TiO_2_ (3)	30	3	24	9	0	3
Geo–rGO–TiO_2_ (0.5, 3)	30	3	24	9	0.5	3
Geo–rGO–TiO_2_ (1, 3)	30	3	24	9	1.0	3

**Table 2 membranes-11-00966-t002:** The physical properties of the resulting Geo–rGO–TiO_2_ nanocomposites depicted in Figure 1.

Sample	Mass (g)	Length (mm)	Outer Diameter (mm)	Thickness (mm)	Density (g cm^−3^)
Geopolymer	28.304	101.1	19.5	2.2	1.04
Geo–TiO_2_ (3)	26.209	100.2	19.6	2.1	1.01
Geo–rGO–TiO_2_ (0.5, 3)	25.426	100.2	19.7	2.3	0.98
Geo–rGO–TiO_2_ (1, 3)	26.414	100.1	19.5	2.2	0.97

**Table 3 membranes-11-00966-t003:** Atomic ratio of Si:Al and Na:Al at the outer layers of the samples, in at.%.

Sample	Si	Al	Na	C	Ti	Si:Al	Na:Al
Geopolymer	53.33	28.84	17.83	0	0	1.85	0.62
Geo–TiO_2_ (3)	46.91	28.00	13.44	0	11.64	1.67	0.48
Geo–rGO–TiO_2_ (0.5, 3)	16.06	11.33	8.23	3.61	4.46	1.42	0.73
Geo–rGO–TiO_2_ (1, 3)	3.86	2.77	28.82	6.48	1.04	1.39	10.40

**Table 4 membranes-11-00966-t004:** Atomic ratios of Si:Al and Na:Al at the samples cross-sections, in at.%.

Sample	Si	Al	Na	C	Ti	Si:Al	Na:Al
Geopolymer	52.38	29.45	18.17	0	0	1.78	0.62
Geo–TiO_2_ (3)	16.89	11.93	8.57	4.23	0	1.42	0.72
Geo–rGO–TiO_2_ (0.5, 3)	14.65	9.60	9.05	2.79	9.22	1.53	0.94
Geo–rGO–TiO_2_ (1, 3)	14.05	10.32	4.55	3.20	11.30	1.36	0.44

**Table 5 membranes-11-00966-t005:** BET parameters for pervaporation membrane samples.

Parameter	Geopolymer	Geo–TiO_2_ (3)	Geo–rGO–TiO_2_ (0.5, 3)	Geo–rGO–TiO_2_ (1, 3)
Surface area (m^2^ g^–1^)	26.031	35.590	29.573	25.549
Total pore vol (cc g^–1^)	0.2020	0.1583	0.1787	0.1666
Average pore diameter (nm)	15.52	7.99	12.90	13.40

## Data Availability

Not applicable.

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
