# Peer review of "Pervaporation Membranes for Seawater Desalination Based on Geo–rGO–TiO2 Nanocomposites. Part 1: Microstructure Properties"

_membranes, 2021, doi:10.3390/membranes11120966_

Round 1
Reviewer 1 Report
In this paper, Geo-rGO-TiO2 Nanocomposites was prepared and its microstructure properties were systematically studied. Article content is substantial, data is complete. Some questions are as below:
- The sentence “The presence of unreacted metakaolin can be seen in sample geopolymer (Figure 10(a))” in line 289, which part is unreacted metakaolin in Figure 10(a) ?
- In the line 308 to 311, the sentence “the rGO nano-sheets and TiO2 nanoparticles cover a lot or areas” only proves that the filler successfully blends into the substrate, but does not prove that the filler plays a major role in the process of seawater desalination because there is no pervaporation desalination property data of Geo-rGO-TiO2 Nanocomposites membrane.
- Can Geo-rGO-TiO2 Nanocomposites membrane still reject salt for its average pore diameter reach to ~13nm by BET test?
- The mass transfer driving force of pervaporation is the saturated vapor pressure difference between the upper and lower sides of the membrane, and the feed does not need pressure. When the Geo-rGO-TiO2 composite membrane is tested for high pressure pervaporation (line 367), will the desalination process become a reverse osmosis or nanofiltration process?
Author Response
Response to Reviewer 1 Comments
Thank you for your insightful comments and suggestions on the paper title: Pervaporation Membranes for Seawater Desalination Based on Geo-rGO-TiO2 Nanocomposites. Part 1. Microstructure Properties. We have gone through all the comments critically and incorporated the changes in the revised manuscript accordingly.
Please refer to the list below to see the author’s response to the comments and suggestions provided.
Point 1: The sentence “The presence of unreacted metakaolin can be seen in sample geopolymer (Figure 10(a))” in line 289, which part is unreacted metakaolin in Figure 10(a)?

Response 1: In the revised manuscript, the unreacted metakaolin spots in Figure 10(a & a') have been identified as red circles and explained as follows (line 437-440):
The presence of unreacted metakaolin can be seen in the sample geopolymer, as indicated by the red circle (Figure 10(a & a’)). The existence of unreacted metakaolin in the geopolymer matrix has a substantial impact on the mechanical strength, pore distribution, and size of the resultant nanocomposite [32,49], and hence the overall quality of the Geo-rGO-TiO2 nanocomposite as a membrane.
Point 2: In the line 308 to 311, the sentence “the rGO nano-sheets and TiO2 nanoparticles cover a lot or areas” only proves that the filler successfully blends into the substrate, but does not prove that the filler plays a major role in the process of seawater desalination because there is no pervaporation desalination property data of Geo-rGO-TiO2 Nanocomposites membrane.
Response 2: Thank you for this comment. We have revised the paragraph by adding the following sentence (line 498 – 501):
The distribution of rGO and TiO2NPs on the surface and cross-sections of geopolymer is one of the prerequisites if the composite is to be applied as a pervaporation membrane in the process of seawater desalination, with rGO serving as nano-channels for water molecules and TiO2NPs acting as an anti-fouling material [9,10,17].
Point 3: Can Geo-rGO-TiO2 Nanocomposites membrane still reject salt for its average pore diameter reach to ~13 nm by BET test?
Response 3: This is very important question, and it became the main aspect of our effort to improve the performance of Geo-rGO-TiO2 nanocomposite as a membrane. Thank you, and our response has been included in the revised manuscript as follows (line 670 – 678):
Because the average pore size of the Geo-rGO-TiO2 nanocomposite formed is roughly 13 nm and the average of pore volume is about 0.1764 cc g-1, the percentage of salt rejection will be quite low. The preliminary test conducted to examine the ability of the prepared membranes to reject salt during the pervaporation process revealed that at 60 °C feed temperature, the drop in salinity of saltwater with an initial salinity of 28.6 ppt was only about 30 percent, with a flux of around 2.75 kg m-2 h-2. The results suggest that the selectivity of the membrane should be improved by reducing the pore size and total pore volume of the Geo-rGO-TiO2 nanocomposite. This can be achieved by solidifying the geopolymer paste through longer curing time and increasing the weight percentage of rGO and TiO2NPs filler.
Point 4: The mass transfer driving force of pervaporation is the saturated vapor pressure difference between the upper and lower sides of the membrane, and the feed does not need pressure. When the Geo-rGO-TiO2 composite membrane is tested for high pressure pervaporation (line 367), will the desalination process become a reverse osmosis or nanofiltration process?
Response 4: This is another important point. Our response in the revised manuscript is (line 679-685):
In a pervaporation process of seawater, the feed side of the membrane is directly contacted with the saturated liquid while the permeate side is kept at a vacuum to maintain the chemical potential difference for water to diffuse into nano-pores across the membrane thickness, similar to the transport mechanism of a nanofiltration membrane. In this case, the pervaporation membrane's mechanism is a phase change process, and unlike reverse osmosis, it does not need to overcome the osmotic pressure of salt water [55,56].

Reviewer 2 Report
1. The elemental compositions of the starting materials were adjusted to have a given atomic ratio. This was the same composition as previously reported. How would the change in the atomic composition affect the material? In other words, is the materials fabrication sufficiently robust to produce the same material with small deviation in the initial composition?
2. Why not use mica? Mica is a natural 2D material. It could be used to replace kaolin or GO in the presented work. An explanation for the choice of materials and their benefits over existing alternatives should be given in the manuscript in more details.
3. Some critical discussions should be added throughout the manuscript highlighting the drawbacks and limitations of the developed membranes.
4. Cross-section of the membranes is crucial to be reported. It is an essential feature for all membranes reported in the literature. Show SEM for the top layer as well as across the whole of the membrane.
5. The tensile strength reported has no errors. The authors should measure all the mechanical properties of the membranes using independently prepared samples, report the averages and add error bars as standard deviations. Tensile strength on its own does not suffice.
6. Natural binder for GO membranes is an interesting concept with scarce reports so far, an emerging area, and therefore prior-art should be briefly mentioned in the introduction (10.1016/j.mtchem.2021.100602; 10.1007/s11595-018-1924-7).
7. Section 2.3 is unnecessary because the authors did not synthesis the TiO2 NPs contradicting the subheading, but they purchased it. Simply mention under the materials section where the NPs were obtained from.
8. The thickness and water contact angle of the membranes should be reported and also compared with the literature. The results should be placed into context in terms of application: what are the requirements of the intended application? Some further explanation on this is necessary.
9. Figure 1 should be provided in a way that the actual scale is legible. Currently the numbers cannot be read and the scale is unclear, which is unacceptable.
10. TiO2 NPs were used to cover the membrane and decrease biofouling. What was the rationale for the selection of these NPs? Are they environmentally friendly? There are alternative natural materials, such as polyols e.g. polydopamine that could be used for the same purpose and should be referred to (10.1016/j.memsci.2020.118881; 10.1016/j.memsci.2020.118007).
11. Comparison with the literature should be provided at the end of the results section. What has been achieved? How does the material compare with similar materials in the literature? What is unique for the new materials? How does the mechanical strength compare with other membranes for the same target application?
12. Both the quotient (“x/y”) and negative exponent (“x y-1”) formats are used in the manuscript for units. Either of them should be used consistently, preferably the negative exponent format, which is recommended by the IUPAC.
13. The captions are very short and not informative. Elaborate more to facilitate understanding.
14. The conclusion section is short and vague. Elaborate more on the main research findings, and mention the most important values, i.e. summarize the most important research findings in a quantitative way. Novelty and potential impact should be mentioned.
Author Response
Response to Reviewer 2 Comments
Thank you for your insightful comments and suggestions on the paper title: Pervaporation Membranes for Seawater Desalination Based on Geo-rGO-TiO2 Nanocomposites. Part 1. Microstructure Properties. We have gone through all the comments critically and incorporated the changes in the revised manuscript accordingly.
Please refer to the list below to see the author’s response to the comments and suggestions provided.
Point 1: The elemental compositions of the starting materials were adjusted to have a given atomic ratio. This was the same composition as previously reported. How would the change in the atomic composition affect the material? In other words, is the materials fabrication sufficiently robust to produce the same material with small deviation in the initial composition?
Response 1: Thank you for this very important point in the synthesis of geopolymer. The following explanation has been added in the revised manuscript (lines 290–302).
The synthesis of geopolymer involves the dissolution of mineral aluminosilicates (such as metakaolin, fly ash, and slag furnace) in an alkaline solution, hydrolysis of Al3+ and Si4+ components, and condensation of specific aluminate and silicate species. In alkaline conditions, the resultant of Al atom is determined by aluminate species of [Al(OH)4]-, whereas Si atom is determined by silicate species of [SiO(OH)3]- and [SiO2(OH)2]- to [SiO(OH)3]- increasing with pH value. The setting and hardening of geopolymer take place as a result of condensation between aluminate and silicate species. The experimental results, such as the one reported by Weng & Sagoe-Crentsil [36], indicate that the condensation between aluminate and silicate species occurs more rapidly than that between silicate species themselves, but the exact mechanism involved is not fully understood yet. This condition, together with the presence of impurity minerals in the starting materials, will result in a minor variation from the calculated compositions of the geopolymer final product.
Point 2: Why not use mica? Mica is a natural 2D material. It could be used to replace kaolin or GO in the presented work. An explanation for the choice of materials and their benefits over existing alternatives should be given in the manuscript in more details.
Response 2: Thank you for this question and suggestion. It is worth considering for our next study. Our response is as follows.
First, the use of the starting materials; kaolin, rGO and TiO2NPs, in this study is based on our extended work from a previous study (2019), a project that was financed by the Ministry of Education and Culture, Republic of Indonesia, and hence cannot be changed by other materials that have been proposed and agreed upon. The rationale for the usage of kaolin, rGO, and TiO2NPs has been mentioned in the introduction of the original manuscript.
Second. We are not aware that mica can be used as a precursor of geopolymer synthesis or to replace GO as a two-dimensional (2D) material with similar physico-chemical properties.
Point 3: Some critical discussions should be added throughout the manuscript highlighting the drawbacks and limitations of the developed membranes.
Response 3: Thank you for these suggestions We have included some critical discussions about the developed and the current membrane in the revised manuscript as follows:
- (line 79 – 86) “The selection of materials and preparation processes for producing organic or inorganic membranes for seawater desalination have been the subject of extensive investigation. However, due to the inherent features of the material, the trade-off between membrane selectivity-flux, biofouling deposition on the membrane surface, membrane durability and lifespan, and energy consumption, the majority of them do not meet the intended performance [6,7]. The goal of this research is to use inorganic nanocomposites comprised of geopolymer as a binder, reduced graphene oxide (rGO), and TiO2NPs as fillers to solve the aforementioned shortcomings of existing saltwater desalination”.
- (line 437 – 440) The existence of unreacted metakaolin in the geopolymer matrix has a substantial impact on the mechanical strength, pore distribution, and size of the resultant nanocomposite [32,49], and hence the overall quality of the Geo-rGO-TiO2 nanocomposite as a membrane”.
- (line 595 – 601) “Geopolymer is a porous cementitious material with a wide range of pore sizes and other voids, such as entrapped and entrained air and capillary pores. When a geopolymer's surface is exposed to water, it absorbs most of the water due to its high hydrophilicity. This reduces the material's lifetime and causes a problem with durability. As a result, to function as a good membrane, the geopolymer surface must be water-resistant. If the geopolymer's hydrophobicity is increased, the rate of water penetration through the material will be slowed”.
- (line 666 – 674) “Because the average pore size of the Geo-rGO-TiO2 nanocomposite formed is roughly 13 nm, the percentage of salt rejection will be quite low. The preliminary test conducted to examine the ability of the prepared membranes to reject salt during the pervaporation process revealed that at 60°C feed temperature, the drop in salinity of saltwater with an initial salinity of 28.6 ppt was only about 30 percent, with a flux of around 2.75 kg m-2 h-2. The results suggest that the selectivity of the membrane should be improved by reducing the pore size and total pore volume of the Geo-rGO-TiO2 This can be achieved by solidifying the geopolymer paste through longer curing time [49], and increasing the weight percentage of rGO and TiO2NPs filler”.
Point 4: Cross-section of the membranes is crucial to be reported. It is an essential feature for all membranes reported in the literature. Show SEM for the top layer as well as across the whole of the membrane.
Response 4: Thank you for this suggestion. The SEM images for the outer layers as well as the cross-sections of each sample have been added with two different magnifications .
Point 5: The tensile strength reported has no errors. The authors should measure all the mechanical properties of the membranes using independently prepared samples, report the averages and add error bars as standard deviations. Tensile strength on its own does not suffice.
Response 5: Thank you for this suggestion. The splitting tensile measurements for all prepared samples have been completed. The average and the standard deviations of the tensile strength are shown in Figure 15.
Point 6: Natural binder for GO membranes is an interesting concept with scarce reports so far, an emerging area, and therefore prior-art should be briefly mentioned in the introduction (10.1016/j.mtchem.2021.100602; 10.1007/s11595-018-1924-7).
Response 6: Thank you for this suggestion. The use of natural binder for graphene-based material has been added into the revised manuscript (line 98 – 108).
The usage of graphene-based material as a filler in membrane manufacturing with organic polymers such as polyimide and polylactic acid (PLA) as a binder has been recognized [11,12]. The organic binder was discovered to have excellent flowability, homogeneity, and to establish a strong chemical bond with GO or rGO. More Recently, a nanofiltration membrane with excellent thermo-mechanical properties and stable performance for water or acetone filtration was reported using rGO as a two-dimensional (2D) material combined with one-dimensional (1D) NaFe2S or NFS to form a ternary composite by using silkworm pupae protein as an organic binder [13].
Point 7: Section 2.3 is unnecessary because the authors did not synthesis the TiO2 NPs contradicting the subheading, but they purchased it. Simply mention under the materials section where the NPs were obtained from.
Response 7: Thank you for this comment. Section 2.3 has been deleted from the manuscript, and the procurement of TiO2NPs is mentioned under the materials section.
Point 8: The thickness and water contact angle of the membranes should be reported and also compared with the literature. The results should be placed into context in terms of application: what are the requirements of the intended application? Some further explanation on this is necessary.
Response 8: Thank you for this suggestion. The physical properties of the Geo-rGO-TiO2 nanocomposite have been listed in Table 2. The water contact angle measurement results have been included in section 3.7 of the manuscript and discussed accordingly. We must admit that the commercial apparatus for measuring the water contact angle is not available in our lab or in any other labs we are aware of, therefore we build our own.
Point 9: Figure 1 should be provided in a way that the actual scale is legible. Currently the numbers cannot be read and the scale is unclear, which is unacceptable.
Response 9: Figure 1 has been replaced and the scale should be able to read.
Point 10: TiO2 NPs were used to cover the membrane and decrease biofouling. What was the rationale for the selection of these NPs? Are they environmentally friendly? There are alternative natural materials, such as polyols e.g. polydopamine that could be used for the same purpose and should be referred to (10.1016/j.memsci.2020.118881; 10.1016/j.memsci.2020.118007).
Response 10: Thank you for this question and suggestion. It is worth considering for our next study. Our response is as follows.
First, the use of TiO2NPs as anti-biofouling in this study based on our extended work from previous study, a project that was financed by the Ministry of Education and Culture, Republic of Indonesia and hence we cannot change the materials that has been proposed and agreed.
Second, the rationale for the selection of TiO2NPs as an anti-fouling agent is already mentioned in introduction part lines 109–120 of the original manuscript.
Third, the primary research interest of our group is in inorganic polymers (geopolymers) based on aluminosilicate minerals, particularly the use of geopolymer as a new functional material beyond its cementitious properties, and therefore, we have minimal expertise with organic polymers as a binder or anti-fouling substance (such as polydopamine (PDA)). As a result, we do not discuss this type of material in detail as we have very little knowledge about it. The suggested articles were referred to as follows (line 120 – 125):
The anti-fouling properties of TiO2NPs could also be obtained from organic polymers such as polydopamine (PDA) coated on a nanocomposite of polybenzimidazole (PBI)-GO membrane, as well as a mixed-membrane of cellulose acetate-polydopamine-sulfobetaine methacrylate (P(DA-SBMA)) nanoparticle for oil-in-water separation [19,20]. It has been reported that a combination of GO and PDA works well as a membrane anti-fouling agent.
Point 11: Comparison with the literature should be provided at the end of the results section. What has been achieved? How does the material compare with similar materials in the literature? What is unique for the new materials? How does the mechanical strength compare with other membranes for the same target application?
Response 11: Thank you for this suggestion. A paragraph has been added at the end of the results section as follows (744 – 758):
The hollow pervaporation membrane for seawater desalination fabricated by using geopolymer-based material was reported by He et al. [29]. The authors intentionally transformed the geopolymer into self-supporting zeolite as a membrane with a thickness of approximately 10 mm, a compressive strength of around 57 MPa, a micropore volume of around 0.049 cc g-1, and achieved 99.5% sodium ion rejection at a temperature of 90°C.
The features of the geopolymer membranes described here are different. The Geo-rGO-TiO2 nanocomposite membranes produced in this study had an average micropore size of 0.1764 cc g-1, a thickness around 2.00 mm, and a splitting tensile strength of 0.35 MPa. The preliminary test of the produced membranes showed that the salt rejection was about 30% and the water flux was around 2.75 kg m-2 h-2 at a feed temperature of 60°C. The results suggest that the membranes developed for desalination of saltwater are still ineffective. In order to function as an excellent seawater desalination pervaporation membrane, it is necessary to fine-tune the geopolymer microstructure, pore size and total pore volume, membrane thickness, and the amount of rGO and TiO2 NPs included into the geopolymer network.
Point 12: Both the quotient (“x/y”) and negative exponent (“x y-1”) formats are used in the manuscript for units. Either of them should be used consistently, preferably the negative exponent format, which is recommended by the IUPAC.
Response 12: Thank you for this suggestion. We have replaced the writing of the units as recommended by IUPAC.
Point 13: The captions are very short and not informative. Elaborate more to facilitate understanding.
Response 13: All the captions of the figures in the manuscript have been elaborated. Thank you for this suggestion.
Point 14: The conclusion section is short and vague. Elaborate more on the main research findings, and mention the most important values, i.e. summarize the most important research findings in a quantitative way. Novelty and potential impact should be mentioned.
Response 14: Thank you for this suggestion. The conclusion has been rewritten as follows:
“A ternary nanocomposite based on geopolymer, rGO and TiO2 was successfully synthesized as a potential inorganic pervaporation membrane for seawater desalination. The physico-chemical properties of the fabricated metakaolin-based geopolymer, rGO, and TiO2 nanoparticles influence the microstructure and strength of the manufactured Geo-rGO-TiO2 nanocomposite. The addition of TiO2NPs and rGO initiated the crystalline phase of the geopolymer matrix, and enhanced the mechanical strength of the membranes. The highest tensile strength of the membrane is about 0.35 MPa and it is able to withstand the feed pressure of up to 3 Pa. The Geo-rGO-TiO2 nanocomposite roughly fulfill all of the parameters for a pervaporation membrane, as measured by pore surface area, pore size, and distribution. The mean pore size was found to be around 13 nm, and the pore volume was found to be around 0.1764 cc g-1, both of which must be reduced for the nanocomposite to function as an effective membrane for seawater desalination. The addition of rGO and TiO2NPs to the geopolymer increases its hydrophobicity, resulting in a water contact angle of up to 63o, that would prevent excess seawater from permeating the membrane during the pervaporation process. The presence of rGO nano-sheets on the surface and in the bulk of the geopolymer will also function as nano-channels, allowing water to freely flow out of the membrane while rejecting salt. The TiO2NPs' superior qualities will aid the geopolymer matrix in improving the membrane's anti-biofouling properties. The results of this study suggest that Geo-rGO-TiO2 nanocomposite is a new functional inorganic material that has a lot of potential for application as a pervaporation membrane for seawater desalination because all of the initial components are widely available and inexpensive”.

Round 2
Reviewer 1 Report
- The Introduction can be further simplified.
- Combining this paper(Part 1) and your second paper(Part 2) into one paper may make your paper fuller.
Reviewer 2 Report
The comments have been addressed and the manuscript has been thoroughly revised.